# Decolonising Fire Science by Reexamining Fire Management across Contested Landscapes: A Workshop Approach

Abigail Rose Croker [1,2,3,*], Adriana E. S. Ford [2,4,5], Yiannis Kountouris [1,2], Jayalaxshmi Mistry [2,6], Amos Chege Muthiuru [2,5], Cathy Smith [2,6], Elijah Praise [7], David Chiawo [8] and Veronica Muniu [8]

1. Centre for Environmental Policy, Imperial College, London SW7 1NE, UK
2. Leverhulme Centre for Wildfires, Environment and Society, London SW7 2AZ, UK; a.ford@imperial.ac.uk (A.E.S.F.); j.mistry@rhul.ac.uk (J.M.)
3. Grantham Institute, Imperial College, London SW7 2AZ, UK
4. Department of Life Science, Imperial College, London SW7 2AZ, UK
5. Department of Geography, King's College, London WC2B 4BG, UK
6. Department of Geography, Royal Holloway University of London, London TW20 0EX, UK
7. Department of Biological Science, University of Nairobi, Nairobi 30197-00100, Kenya
8. Centre for Biodiversity Information Development, Strathmore University, Nairobi 59857-00200, Kenya
* Correspondence: a.croker20@imperial.ac.uk

**Abstract:** In many landscapes worldwide, fire regimes and human–fire interactions were reorganised by colonialism and continue to be shaped by neo-colonial processes. The introduction of fire suppression policies and state-centric property-rights systems across conservation areas and the intentional erasure of Indigenous governance systems and knowledge have served to decouple Indigenous fire-dependent communities from culturally mediated fire regimes and fire-adapted landscapes. This has driven a decline in anthropogenic fires while simultaneously increasing wildfire risk where Indigenous people have been excluded, resulting in widespread social–ecological vulnerabilities. Much contemporary fire research also bears colonial legacies in its epistemological traditions, in the global geographical distribution of research institutions, and the accessibility of research outputs. We report on a two-day workshop titled 'Fire Management Across Contested Landscapes' convened concurrently in Nairobi, Kenya, and London, UK. The workshop formed part of a series of workshops on 'Decolonising Fire Science' held by the Leverhulme Centre for Wildfires, Environment and Society, UK. The workshop in Nairobi invited diverse Kenyan stakeholders to engage in participatory activities that facilitate knowledge sharing, aiming to establish an inclusive working fire network. Activities included rich pictures, world café discussions, participatory art, and the co-development of a declaration to guide fire management in Kenya. Meanwhile, in London, Leverhulme Wildfires researchers explored participatory research methodologies including rich pictures and participatory video, and developed a declaration to guide more equitable research. There were opportunities throughout the workshop for participants in Nairobi and London to engage in dialogue with one another, sharing their experiences and understandings of complex fire challenges in Kenya and globally.

**Keywords:** fire management; fire governance; participatory approaches; decolonisation; cultural burning; wildfires; protected areas; conservation; intercultural; Kenya; colonialism; resource management





## 1. Introduction

Fire is an essential disturbance event that has shaped the evolution of savanna social–ecological systems across Sub-Saharan Africa. Since as early as 1.5 million years ago, humans have used fire for survival, spiritual purposes, and to achieve a range of agricultural and pastoralist livelihood objectives [1,2]. These practices have shaped the vegetational composition, abundance, and distribution of highly biodiverse and culturally rich African savannas [3,4]. Maintaining equilibrium between human fire activity and

ecological stability across the region is challenging. Complex geopolitical histories, climate change, economic transitions, population growth, and rural–urban migration challenge long-established fire regimes [5]. In recent years, the pyro-geographic landscape across Africa has shifted, revealing historical inequalities in resource management where the interactions between fire, environment, and society have been disrupted [6–8]. Where colonial administrations introduced fire suppression policies and displaced Indigenous and local fire-user communities from their ancestral lands, a coloniality–wildfire nexus has emerged. Inside protected areas, the accumulation of flammable biomass has driven an increase in extreme wildfire events [9], while outside park boundaries, population growth and sedentarisation have accelerated bare ground cover, reducing forage and pasture availability [10]. Fire suppression policies condemning local burning practices have been reinforced through post-independence and international institutions [11,12] and perpetuated through the media and conservation–development agencies [13]. This continues to inform exclusionary management and policy interventions [14].

Since the mid-twentieth century, scientists and practitioners have increasingly recognised the regenerative role of fire in fire-dependent and fire-adapted ecosystems. This has resulted in the implementation of prescribed early dry season burning in some parts of the world [6,15], for example, in "Indigenous-led" savanna-burning emissions abatement schemes in Australia [16–19], and in many cultural burning projects in the USA [20–22]. However, most of the research informing early burning practices is based on field plot experiments with little or no human presence [3,23,24], rather than Indigenous peoples' interannual fire practices [25]. The exclusive prescription of early burning in fire-dependent savanna woodlands and fire-adapted forests can alter ecosystem functioning and reduce carbon sequestration in the long-term where some high-intensity fires are necessary components of the systems' adaptive cycle [16].

To date, Indigenous-rights discourse in fire management has mainly been constructed by Western researchers, or "outsiders", in the local context in which they are researching [26]. Our knowledge of Indigenous fire use is mostly underpinned by subjective interpretations in the published literature, often founded on brief superficial encounters between the researcher and researched [27]. These challenges are epitomised in disaster research in the non-Western world, with those who are most vulnerable to disasters and experiencing their impacts firsthand often being marginalised in their reporting and understanding [28,29]. Disaster research is dominated by Western or "outside" scholars, often driven by media and politics, and frequently lacking knowledge of the affected area and local cultures. This can create misconceptions of resilience and vulnerability in the local cultural context and remove our understanding of disaster events from their original political agenda [13]. For example, international response to disasters often imposes neo-colonial action in local settings where fixed vulnerability indices and assessment frameworks are used to inform risk reduction strategies. This can remove situated political and social dimensions that impact, and are impacted by, the disaster, and that influence capacity to reduce and adapt to risk in the long-term [29]. Additionally, research is often based on hypothesis testing based on a set of predetermined parameters. This can omit multiple system variables at the local level, risking selection, information, and confounding bias [30,31]. In this process, the diverse knowledge systems and evolving practices of Indigenous people are assimilated into a singular epistemic tradition [32], a common cultural, ethnolinguistic, and spiritual representation, from which "an Indigenous knowledge" can be easily extracted [33]. In the absence of ethical and non-extractive research relations with local people, visiting researchers can be at least twice removed from the realities on the ground. Thus, the incorporation of Indigenous people in fire management risks reconstructing institutional and knowledge hierarchies in the reimagination of equitable fire governance.

In recent years, there has been growing recognition of the failings of current fire management approaches across diverse social–ecological contexts. This is driving an ideational shift in disaster action and natural resource management. Researchers working in the field of disaster studies are aware of the Western hegemonies that underpin their

science, stating the need to transfer *"power to local scholars to take the lead in studying disasters"* since they *"have greater first-hand experience of disasters"*. In this way, disaster research can *"inform current policies and practices geared towards reducing the current risk of disaster, as much as to avoiding the creation of new risks in the future"* [28,29,34–36]. More recently, a global jurisprudence in respect of Indigenous rights and empowerment has emerged in international climate politics [14]. This is supported by mounting scientific evidence acknowledging the need for the inclusion of diverse knowledge systems in environmental management to support adaptive and resilient social–ecological systems in the context of ongoing climate variability and change [37,38].

Alerting primarily Western-trained research communities to the risks of neglecting local and Indigenous knowledge is essential for overcoming colonial and neo-colonial influences on land use and fire management [21]. Here, we present the structure and results from a workshop conducted in the UK and Kenya aiming to take initial steps towards facilitating inclusive dialogue on decolonising fire research and establishing mutually beneficial relationships for the incorporation of local and Indigenous knowledge in fire management [26,39,40].

On 1–2 December 2022, the Leverhulme Centre for Wildfires, Environment and Society ("Leverhulme Wildfires"), UK, and the Centre for Biodiversity Information Development (BID-C) at Strathmore University, Kenya, supported by the Grantham Institute, Imperial College London, UK, held a two-day workshop in Nairobi on decolonising fire science and management. The workshop, titled *"Fire Management Across Contested Landscapes"*, was the third workshop in the Leverhulme Wildfires' *Decolonising Fire Science* series and was held in parallel with a training workshop in London.

In London, natural and social fire scientists at Leverhulme Wildfires engaged in training activities and conversations on decolonising research practices. Training activities focused on techniques that can promote equitable and experiential exchange, with the aim of making local and Indigenous voices heard, rather than interpreted through the lens of the Western academically trained researcher [26]. In Kenya, diverse stakeholders and rightsholders living and working across Kenya's contested fire-prone landscapes (including: the Tsavo Conservation Area, Amboseli ecosystem, Maasai Mara National Reserve and Community Conservancies, Mount Kenya Conservation Area, Meru Conservation Area, and Kirisia Forest) were invited to engage in participatory activities that facilitate knowledge sharing, aiming to establish an inclusive working fire network. Activities included rich pictures, world café discussions, and a collaborative art session. These activities were supported by a series of presentations on fire science and management and a talk on *"how to influence policy and draft a white paper from the bottom-up"*. A local artist, Shadrack Musyoki, led the collaborative art session and created one live art piece reflecting the ongoing workshop discussions and one summary piece displaying his own interpretation of the discussions [41].

This workshop provided a novel opportunity for local stakeholders and rightsholders to actively interrogate colonial legacies in fire management and explore normative future scenarios, particularly those that prioritise local and indigenous peoples' decision-making rights and access to benefits in resource governance. To our knowledge, this was the first intercultural workshop to explicitly address fire-related challenges and the failures of current fire suppression and prevention operations in the Kenyan context.

We, the authors of this paper, co-created this workshop to bring together these different groups of people to engage in open dialogue and share perspectives over wildfires. We are of European and Kenyan descent, speak multiple languages, and hold diverse social identities, expressing differences in our positionalities. E. Praise, A. Muthiuru, D. Chiawo, and V. Muniu are Indigenous to Kenya. A. Muthiuru assisted in planning the workshop and attended the workshop in Nairobi as a participant. E. Praise, D. Chiawo, and V. Muniu were involved in the co-creation and facilitation of the workshop. We explicitly acknowledge how levels of positionality within the authorship team (e.g., academic rank, age, gender, ethnicity, political affiliations, and religious beliefs) might have influenced how the workshop was

developed, facilitated, and communicated. In defining ourselves to one another, we celebrated diversity within the authorship team throughout the workshop and in writing this report, explicitly acknowledging how our beliefs, values, and worldviews have been shaped by our personal experiences, geographies, and political and social affiliations. All authors are associated with academic institutions, influencing our epistemological assumptions and how we approach research. Therefore, we gained consent from all participants to share their original words in this report, and they were sent regular drafts to ensure that the authors interpreted their perspectives as intended. They have each been individually thanked in the Acknowledgments section for their participation and ongoing enthusiasm for addressing fire governance challenges in Kenya.

In the following sections, we detail Leverhulme Wildfires' Decolonising Fire Science workshop series, summarising the first two workshops to contextualise the aims, scope, and approach taken in the third workshop. We then report on the third workshop, providing (i) background information on fire management in the context of Kenya's conservation areas, (ii) an outline of the workshop's scope and objectives, (iii) an account of the participatory activities we engaged in and key outputs, and finally, (iv) next steps. This paper contributes to the growing body of literature on the importance of decolonisation in science, management, and disaster studies, equitable collaboration in participatory research and culturally grounded enquiries, and centering local and Indigenous perspectives in environmental stewardship. We hope this paper can provide useful directions for researchers trying to navigate this complex field.

## 2. Decolonising Fire Science Workshop Series

*"Ethical concerns should have the same primacy as research questions"* [28].

In recognition of the challenges outlined in the previous section, Leverhulme Wildfires initiated a Decolonising Fire Science workshop series in April 2022. This series explored opportunities for participatory research approaches that foreground relationality and experience in the construction of scientific understandings of fire-use communities and landscapes [42,43]. Central to this approach were questions of power, justice, legitimacy, and "otherness", such that we sought to understand how our own situational self-identity impacts our interpretation of the "other"—the research participants—identifying how patterns of power and privilege might shape our inquiries in different contexts [44,45]. This series also addressed the ways in which research can reproduce colonial paradoxes in disciplinary power on decolonisation [40,46]. Beyond research, this series investigated the institutional mechanisms and rule-based approaches embedded within international and legally binding agreements that continuously exclude Indigenous and local people from decision-making processes (e.g., deliberation, participation, and conflict) and reinforce power hierarchies between scientific, policy-relevant, and local governance systems over fire use [47,48].

The first two workshops, *"An Introduction"* and *"Critical Conversations"*, introduced questions of decoloniality in research to Leverhulme Wildfires members and facilitated in-depth discussions over how to foster decolonisation in fire research. Both social and natural scientists were encouraged to participate, including individuals working with fire models to whom the conversation might appear less applicable yet who have an important role in promoting equitable fire management through their representation of human–fire interactions. We identified several components of decolonisation that require questioning in our research, as well as challenges that require explicit consideration in future projects to foster the conditions for an equitable fire future. Croker et al. (2022) [49] and Croker and Ford (2022) [50] provide detailed reports on *"Decolonising Fire Science: An Introduction"* and *"Decolonising Fire Science: Critical Conversations"*, respectively. During these workshops, rich pictures, participatory videos, intercultural workshops, and participatory modelling and scenario analysis were identified as useful methods to assist in building intersubjective interpretations of the research context and the co-development of narratives, contributing to more equitable forms of engagement with local and Indigenous people and management

outcomes [51–53]. The discussions also highlighted the importance of coupling archival research with community-centered approaches (e.g., sharing oral histories) to gain a clear understanding of the historical and institutional context [4,54]. This can help identify dominant voices and power hierarchies and assist in effectively facilitating participatory research activities [55].

The promises and limitations of participatory methodologies in decolonising fire science were also discussed in the first two workshops [56]. Critiques of the "participation" paradigm include the ways in which attempts to retrieve the voice of "the other" can inscribe false dichotomies between local or Indigenous knowledge versus scientific or Western knowledge [57–59]. Similarly, it was acknowledged how participatory processes are imbued with power relations between researchers and participants and between participants [43,60,61]. Other barriers to achieving decolonisation in research include funding, time, institutional and academic requirements, and the nature and scale of the research project.

Workshop three, *"Fire Across Contested Landscapes"*, was held in Nairobi and hosted by the Centre for Biodiversity Information Development (BID-C), Strathmore University, Kenya. Prior to the workshop, Leverhulme Wildfires member Abigail Croker had been carrying out fieldwork in Kenya in collaboration with a local researcher, Naftal Kariuki, to explore wildfire challenges across the Tsavo Conservation Area [30]. They worked with a range of Indigenous and local people, governmental and private agencies, and local NGOs. This workshop was largely motivated by the findings of this research and the realised similarities between fire-related challenges in this region and other fire-prone landscapes worldwide. Participants had expressed the need to *"bring scientists, practitioners, and local people together to just share. [. . .] Planning platforms which bring these people together means that they can plan in their own way, [. . .] based on the realities on the ground. [. . .] Then we can bring the government and these people together, to sit at a table and agree on the direction we need to take, a coordinated direction that we have all agreed on. [. . .] At the moment, the problem is inertia to plan"*.

This workshop was a collaboration between Leverhulme Wildfires and the BID-C, aiming to build upon previous discussions and apply decolonising methodologies in practice and meaningfully contribute to ongoing research with locally relevant outcomes [13]. In recognition of Kenya's complex ecosystems and social–cultural diversity, the geographical scope of this workshop extended across the country's fire-prone protected areas. This also increased the workshop's relevance to Leverhulme Wildfires members researching in different global contexts, providing a practical learning experience to engage in participatory methodologies and reflect upon the opportunities and challenges for decolonising fire science in their own projects.

## 3. Case Report: Fire across Contested Landscapes, Kenya

*"Let the field speak to you"* (Invited Researcher)

### 3.1. Fire Management in Kenya's Conservation Areas

Fire and its interactions with multiple ecological and climate variables affect the structure, functioning, and regeneration of Kenya's fire-adapted and fire-resistant savanna and forested ecosystems [62]. In recent history, anthropogenic climate change has intensified and increased the frequency of ENSO events in East Africa, resulting in multi-year droughts interspersed with heavy rainfall and flooding events [63]. Given that East Africa's wildfire seasons are more than a month longer than in the 1980s [64], the rapid accumulation of flammable biomass during wetter periods, largely concentrated in protected areas where fire suppression has been implemented, poses significant social–ecological threats and vulnerabilities [16].

Kenya has a complex colonial history, and neo-colonial influences continue to shape exclusionary conservation policies and fire management across protected area landscapes [65]. Throughout the 1800s the 1900s, the British colonial government introduced a series of Acts

that legally took control of indigenous territories for public and private development and the expansion of hunting blocks (later converted to conservation areas and ranches). For example, the 1894 Land Acquisition Act and 1896 Indian Acquisition Act allowed authorities and private companies to seize land for public purposes and development projects, and the 1897 Land Regulations Act enabled land to be granted to foreigners on a 99-year lease. The 1902 Crown Lands Ordinance (CLO) endorsed land alienation for white settlement and the expansion of hunting blocks. In 1915, the CLO amendment allowed native reserves to be taken over as crown lands. All unoccupied land in Kenya was declared property of the state, and Indigenous people occupying native reserves became tenants-at-will of the crown and could be evicted at any time [66,67]. Today, Indigenous people and local communities continue to be displaced from large swathes of their ancestral lands to make way for wildlife conservation [67], concentrating human and livestock populations outside protected area boundaries, often on communal lands or group ranches already under pressure from population growth and climate change. Incentivised by external agencies to increase benefits from tourism enterprises, group ranches are increasingly dedicating a portion of their land to wildlife conservation and instating fire suppression policies. For instance, the Wildlife Conservation and Management Act (2013) convicts any person who sets fire to vegetation in protected areas [68]. The introduction of the Community Land Act (2016) is contributing to the sub-division of rangelands and ranches to secure access to competitive resources [69]. Near protected areas where existing market and access infrastructure can be utilised for commercial operations, there is growing foreign private investment in large-scale land developments [70].

Land sub-division and "development-driven displacement" are accelerating inter-group conflict over access to declining resources, particularly where Indigenous and local people feel that the government and international agencies prioritise wildlife over local livelihoods. The intentional decoupling of fire from its sociocultural and environment context since the late 1800s has contributed to the loss of traditional fire knowledge and homogenised Kenya's diverse pyro-geographies. Historical fire regimes, characterised by small-scale fires with heterogenous burning objectives and impacts, have been replaced with a singular, dominant fire regime, characterised by high-intensity wildfires concentrated within and around protected areas where fire suppression is enforced [16,71].

### 3.2. Workshop Scope and Objectives

*"Fire Across Contested Landscapes"* had three core aims: (1) to further the decolonisation strategies of Leverhulme Wildfires, (2) establish long-term research partnerships with local institutions, and (3) explore opportunities for decolonising fire management in Kenya. Given the conflictual nature of fire management in Kenya, this workshop was designed to maximise equitable intercultural exchange between researchers, Kenyan stakeholders and rightsholders, and London-based participants. Workshop facilitators (in Nairobi: Abigail R. Croker, Dr Adriana Ford, Veronica Muniu, and Elijah Praise. In London: Dr Jay Mistry and Dr Cathy Smith) considered the legacies of systemic injustice and layered power relations embedded in intercultural settings to ensure that principles of respect, appreciation, and value were maintained throughout the discussions [52,72,73]. Discussions were translated from English to Swahili where necessary.

Individuals interested or involved in fire management were invited from across Kenya, ensuring that both women and youth were represented in this workshop [8]. Core participants were identified by researchers, then a snowball sampling approach was applied to identify rightsholders who are more difficult to communicate with via technological platforms. The workshop was attended by representatives of public, non-governmental, private, and community organisations. These included the African Wildlife Foundation (AWF), Amboseli Ecosystem Trust (AET), Food and Agricultural Organization of the United Nations (FAO), Garissa University, Karatina University, Kenya Forest Service (KFS), Kenya Wildlife Conservancies Association (KWCA), Kenya Wildlife Service (KFS), Kirisia Community Forestry Association, Mt Kenya CFA, Penda Kujua, Sheldrick Wildlife Trust (SWT),

Taita-Taveta Wildlife Conservancies Association (TTWCA), Taita Wildlife Conservancy, Teita Sisal Estate, Tsavo Heritage Foundation (THF), and Wushumbu Conservancy. Recognising the invisible costs of research [50], transport, accommodation, and meals were provided for all.

### 3.3. Kenya Workshop Activities

We drew upon a systems framework for participatory learning to structure session activities and facilitate equal involvement in social and political dialogue amongst participants [37,74,75]. The workshop agenda was co-developed by international researchers from Leverhulme Wildfires and local fire scientists affiliated with the BID-C. The questions asked to guide discussions were adapted from collaborative fieldwork in the Tsavo Conservation Area, where Indigenous people and local communities were asked to identify fire governance challenges across protected area landscapes and future research requirements. While the agenda and questions were predetermined, a flexible and iterative approach was taken throughout the sessions to allow participants to co-create the narrative, framing their own outcomes on the second day and planning next steps and future action.

First, workshop participants were introduced to the aims of Leverhulme Wildfires and the BID-C, as well as the Decolonising Fire Science series. To contextualise the workshop, participants were provided with a summary of the key principles of decolonisation in research and management. This was informed by critical decolonial scholars from the majority world and existing efforts to decolonise research in fire-dependent social–ecological systems globally [53,76]. Over the two days, workshop participants engaged in participatory activities to foster the first three iterations of the systems framework, supported by presentations delivered by international fire scientists on fire ecology and governance across contested landscapes worldwide [41]. Studies have highlighted the challenges encountered by researchers when empirically investigating experiences over contested issues or where taboos are attached to local norms and beliefs [33,77–80]. Tensions between researcher and participant have been likened to Hegel's master–slave dialectic [81,82], suggesting that researchers can impose a form of consciousness in the researched, influencing their attitudes and behaviour towards a certain issue or how they respond to a question (i.e., what they perceive to be the "correct" answer) to avoid penalisation. Therefore, qualitative research can be skewed to reflect the researcher's worldviews and objectives more accurately, reinforcing problems associated with "parachute science" and the embeddedness of colonial biases in knowledge production [46,53,83,84]. Recognising these challenges, presentations were delivered after the first activity, rich pictures, to gain a better understanding of the participants' interpretation of *"what does fire mean to you?"*.

### 3.3.1. Rich Pictures

The first activity invited participants to reflect upon the question *"what does fire mean to you?"*. Rich pictures are an epistemological constructivist approach that acknowledge diversity in our understandings of a specific issue based on previous experiences and background knowledge [85]. Participants were asked to illustrate their individualistic interpretation of the question using drawings, symbols, and text [86]. Rich pictures can be used to transcend language and cultural barriers to communication and enable participants to "surface" latent thoughts, particularly in the presence of perceived or established sociopolitical hierarchies [1,87]. In this way, they can help individuals discuss divergent understandings of and relationships with fire in their landscape and work towards collective action [87,88].

Rich pictures were used as an introductory activity to facilitate an open unstructured conversation between workshop participants who were mostly unacquainted, from different jurisdictions in Kenya, and worked and lived across diverse social, political, and environmental contexts. Participants were asked to randomly select a table in the room, create an individual rich picture on a shared sheet of paper, and discuss their picture with

their neighbours. The groups were heterogenous, and the discussions were courteous, respecting one another's understanding of fire (Figure 1).

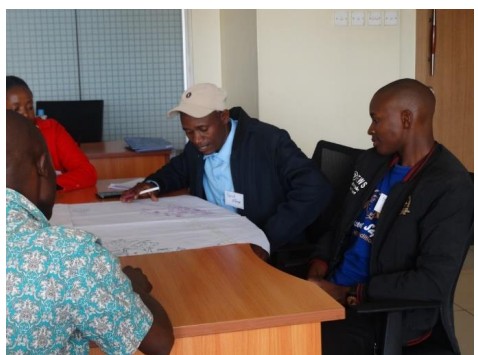 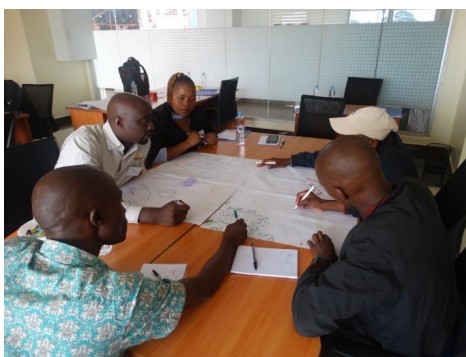

**Figure 1.** Rich picture exercise with Nairobi participants. Original photo: Abigail R. Croker.

Group representatives summarised each table's response to the main question in a plenary session. In their presentations, they revealed both positive and negative attitudes towards fire that were shared between participants due to having experienced similar challenges across fire-prone landscapes. Positive attitudes mostly concerned fire's regenerative role on post-burn vegetation structure and soil composition. However, they explained that this knowledge required a longer-term outlook in grassy ecosystems occupied by both wildlife and livestock due to greater inter- and intraspecific competition:

> *"When there is a fire, vegetation is burned, meaning there is less food for wildlife and livestock. [. . .] At the same time, when there is fire, it produces ash which is a fertiliser. It makes fresh grass grow very fast, which is very beneficial to wildlife and livestock. [. . .] Fires are ecosystem architects; they build forested ecosystems".*

In forested ecosystems, participants found that *"fire is new life"*, creating clearings in the dense canopy and promoting the natural regeneration of species such as *Juniperus procera* and *Hagenia abyssinica* [89]. Some participants attributed charcoal burning, commonly perceived as degradative, to the natural regeneration of Indigenous forests and closed-canopy woodlands, providing it is carried out for subsistence rather than commercial purposes. *Toona ciliata* and *Olea Africana* were identified as benefitting from charcoal burning:

> *". . .fire breaks the dormancy of the seeds. Sometimes [. . .] you have seeds that remain there for 20–30 years. [. . .] Where charcoal has burned and there is good rain", "you find that seedlings grow faster than in other areas, [. . .] and there will be a lot of tree growth".*

Several participants emphasised the importance of subsistence-oriented burning practices for farming, land clearance, grazing, resource harvesting, and survival. They expressed their concerns over the availability of alternative management options in poorer rural regions where access to technology is limited. A few participants also recollected on the centrality of fire in cultural and spiritual beliefs and its role in reinforcing kinship:

> *"We have campfires, where people sit around the fire. It's an ancient tradition. Fire gave us an evolutionary edge; it gave us power over the planet. There are so many hazards and risks with fire, but we were the only ones that harnessed the energy of fire. An equally important thing that fire did was that it forced human bond and interaction. It was around campfires like this, on a starry night. We sat around and our ancestors' shared stories and our children played. For me, it is that positive human fire interaction".*

Participants expressed how local fire use practices have been routinely demonised by state and non-state agents as a leading source of environmental degradation [4]. Negative attitudes towards traditional fire uses are strongly embedded in colonially derived environment policies which have institutionalised fire management at the national level, preventing Indigenous and local people from participating in fire-related decision making [90]. Some senior participants attributed the loss of traditional fire knowledge over the last century

to this process of institutionalisation, expressing feelings of sadness over the growing rift between humans and fire and what this meant for the future of their cultural heritage [54]. For example, they shared sentiments of fire being both feared and revered across conservation landscapes, discussing cases of incendiarism as an indication of agrarian discontent and resistance to colonial land tenure arrangements and fire suppression [91], largely due to observed increases in large wildfires and the perception of socioeconomic risk associated with these events. Wildfires were attributed to local conflict:

> *"Most people use fire as a form of social conflict, [. . .] often between the people living around the protected area and the park authorities. [. . .] A ranger could draw a gun and arrest a community member and they retaliate using fire".*

Observed increases in large wildfire events attributed to social conflict were blamed for instilling anti-fire sentiments within Indigenous and local fire-user communities, largely due to their perceived socioeconomic risk [71]. This positive feedback between agrarian resistance and negative attitudes towards fire across conservation landscapes was highlighted by representatives from community forest associations (CFAs) who joined state officials in condemning community practices:

> *"Charcoal burners go and burn charcoal in the forests and leave the fires burning, often at night"*, and *"men climb trees with fire when they look for honey, using smoke from the fire to smoke out the bees and collect honey, leaving the fire in the tree, burning"*. As a result, they expressed how *"the trees are burning, and animals and people are running"*.

Despite attributing wildfires to their communities, they disagreed with the way government authorities criminalise people for burning, such that *"a community member could be arrested by the KWS for lighting a fire in the park, but I am sure that those KWS members have also lit fires previously"*. They argued that this perpetuates this positive feedback and the strength of colonially inherited understandings of fire. In recent years, growing demands for food, poverty, and unpredictable changes in seasonality have also increased the use of fire as an inexpensive and less labour-intensive tool to *"open the land for cultivation and clear up farms to increase production"*, often burning several times in a single growing season. Historically, agricultural fire use was governed by customary laws, regulating by whom and under what conditions burning can be carried out. However, land sub-division and the breakdown of traditional land governance systems have contributed to uncoordinated burning practices, commonly representative of individualistic ambition rather than collective objectives.

Agricultural fires that escape from local farms into protected areas were repeatedly mentioned as a leading cause of wildfire and of property damage across community lands. Community members were worried about wildfires spreading to their farms, especially in (semi-)arid regions where polyculture agroforestry is practiced to address food insecurity challenges [92,93]:

> *"Projects might fund tree replantation in certain areas. The seeds are replanted and there is a cost for the seedlings and the labour. Then the fire burns all the seedlings and trees already existing in the area. So, money is lost. Trees are lost. And small seedlings that are replanted to improve the degraded area are also burnt".*

Older participants stressed that these challenges will be amplified in the future due to a lack of traditional fire knowledge amongst the younger generations, limited resources to effectively manage burns under worsening wildfire conditions, and increased socioeconomic insecurities under climate change. This reflects an increase in resource-based conflicts across Kenya that have led to displacement and hardship [94]. These negative experiences of fire, occurring alongside multiple social–ecological compound pressures, are reinforcing anti-fire rhetoric, now also perpetuated by individuals within Indigenous and local communities [95].

Participants working for government agencies and international development organisations also attributed attitudinal changes and the ongoing transition from collective to individualistic ambition to macroeconomic influences that are often *"overlooked when*

*talking about fire management"*. At the local level, *"charcoal burners are going into protected areas, perhaps where neighboring communities are working towards economic progress [. . .]. And because of population growth, they are moving into and clearing forests to make room for agriculture. Clearing the forests and bushes using fire"*.

At the regional and national level, *"fire is used in industries which have huge impacts over local areas"*, referring to Kenya's growing commercial biofuel industry which depends on forests for woody biomass, resulting in widespread deforestation and ecological degradation [10,96]:

> *"We get fires where wood is used in industries, but there is also the economic benefit of these industries in terms of manufacturing. So, there is that critical borderline between destruction and economic benefits. The benefits are accrued through production, [. . .] so there is that component of increasing production and using fire, and not destroying the forests where everything is being harvested"*.

One participant emphasised the interconnectedness of local and national energy demands, drawing attention to the increasing deficit between demand and supply for biofuels, particularly on account of rapid population growth. Considering the government's ban on harvesting wood in protected areas, the participants expressed concern over the escalating price of charcoal and subsequent increases in illegal and unsustainable fire use [97,98].

Representatives from conservation organisations and local communities expressed concern over the threat wildfires pose on biodiversity, particularly when *"animals escape from the protected area [. . .] to escape the fires"* and the cascading effects this has on human–wildlife conflict. The concern expressed by community members was partly driven by their intrinsic concern for wildlife, especially amongst pastoralists who highlighted their longstanding co-existence with wildlife. However, they shared how much of these positive interactions had changed around protected areas where, due to declining pasture availability and land fragmentation, they had settled and adopted mixed crop–livestock systems. Participants from different regions shared their experiences of increasing incidences of depredation of livestock by large carnivores, synonymous with observed declines in wildlife inside protected areas where *"after the fires, the water dries up and the animals, because of a lack of water and vegetation, leave"*. Agriculturalists appeared more worried over the threat of crop raiding by elephants and other large herbivores when they *"run away from their habitats to escape the fire"*, destroying farms and jeopardising livelihoods in a single raid [99]. These challenges have increased the use of fire to drive herbivores off crop fields, predators away from livestock, and guard homesteads, increasing the risk of wildfires. Representatives from conservation organisations were more concerned with the effects of wildfire on wildlife populations, as *"elephants are trying to escape, [. . .] giraffes are running for safety, and we have small animals that are burned out, like tortoises and snails. There are dead animals because of fire"*. They noted how *"the fires start in the conservancies, and animals are running towards human settlements, making them vulnerable to poaching"*.

The discussion concluded with participants sharing their views on fire management across the landscape, revealing areas of convergence and divergence between individuals and social–cultural and socioeconomic groups (e.g., different Indigenous and local community groups, conservation NGOs, and development agencies) [95]:

> *"Fire can also be a solution to a problem. When people are fighting for grass, you just burn it, and everyone lives in peace. Burning the common resource stops people fighting over it." "People argue about fire because of how different groups perceive it—positive and negative. Grass is a common resource." "Burning the resource being fought over can result in further conflict. People and their livestock rely on this grass. Where will they find grass to graze on? That is a huge economic loss"*.

> *"You need communities on side. This [. . .] mentality is dangerous—when you position a person's rights aligned with an elephant. When I look at fire governance, I am looking at it from a conservation aspect. Those who are more trained think like the west, and*

*they have authority. There is no local community knowledge in societies [. . .] because it's going in a non-indigenous direction. [...] Fire is a problem to the locals too. Including locals is a solution. The other problem is that the fires lit out of conflict are not intended for wildlife, but they affect them. We need to look for solutions. A transferable model. You solve it here; it needs to be transferred elsewhere".*

3.3.2. World Café Discussion

Participants engaged in world café conversations to network, discuss experiential knowledge, access collective understanding and intelligence, and explore new insights [37,100]. World café centralises free and equal participation in deliberative dialogue in its approach to support locally meaningful action [101,102]. Conversations explored the following questions: "how is fire governed across your landscape?", "what do you want to change about fire management?", "what are the challenges preventing local community involvement in fire governance?", "what are the national and regional barriers to fire management? (e.g., social, economic, political, legal)?", and "what can be done to facilitate equitable fire governance and management?". Each question was presented on a different table, and participants were encouraged to move between tables to engage with each question and comment on the responses already shared by others. The group reconvened to discuss their responses, revealing disparities in their awareness of national fire legislation, and intergenerational differences in their understanding and acceptance of traditional knowledge in fire management.

Multiple management interventions used to prevent and suppress fires were identified by participants, including firebreaks, firefighting teams, surveillance in protected areas, sensitisation meetings with local communities, and resourcing from NGOs for fire control. However, participants agreed that most of these interventions were under-funded, inadequately resourced, and lacked consistency across sectors, land tenure arrangements, and ecosystems, such that *"every sector [is] addressing fire on its own, for example the wildlife and forest sectors. You do not have that integration":*

*"We have committed many resources into fighting fires across the landscape. But the biggest challenge we have is identifying how the fires started and who is doing this. We need to work more on this intelligence aspect so we can identify exactly who is doing what. [. . .] Other areas we have exhausted—we have firebreaks, we have guys working. But when a fire spreads from a neighboring farm into the park, we don't know who started it. Or when a poacher goes into the park and starts a fire".*

According to a KFS officer's experiences, controlled burning used to be practiced as a forest management tool, *"but due to a lack of maintenance of fire breaks and a lack of policy, this does not occur. This lack of this policy means a lack of burning. There is also a funding issue"*. These challenges were attributed to the lack of a fire governance framework in Kenya, or otherwise a lack of awareness of it being addressed in relevant policies. Participants continuously reinforced the point that *"each sector are producing their own strategies. Currently, there is no coordinated policy to manage wildfires"*. Adding to this, they were uncertain over (i) the policy area in which fires are handled (e.g., different resource sectors), (ii) the geographical scale of governance interventions (e.g., based on biogeophysical conditions or jurisdictional boundaries), (iii) who is responsible for making rules on fire use and the conditions under which rules-in-use are enacted across multiple scales (e.g., local, landscape-scale, regional, national) and property-rights systems (e.g., public, private, common-pool resource), and (iv) the repertoire of strategies, shared norms, and rules available to govern fire [14,103,104].

Participants highlighted the complex history and embeddedness of fire regulations in land management policies first established by the British colonial administration. For example, one forest officer referenced the Grass Fires Act (2012) first enacted in 1942 following the Control of Grasses Fires Ordinance (1941) [105,106], highlighting its clear presentation of vegetation burning regulations on privately owned and communal land and on lands adjacent to national reserves. However, the officer noted that the Agricultural, Fisheries, and Food Authority Act (2013) repealed the Grass Fires Act [107], as well as

the Agriculture Act (1955, Rev. 2012) which includes some of its own rules for burning agricultural lands [108]. At the same time, the Wildlife Conservation and Management Act (2013) and Forest Conservation and Management Act (2016) impose strict bans on fire use within and surrounding forests and protected areas [68,109]:

> *"Every sector has a law. The Forest Act, the Grasslands Act, the Wildlife Conservation Act..., each imposing penalties for people burning vegetation. [...] But there is no permission for fire use in the law for the time being. We have it in our plans, in our technical reports. Burning is there, like early burning, but it is not practiced".*

Additionally, *"Most people are not aware of these laws"*, and where land use and tenure systems overlap, are contiguous, or have undergone conversion since these acts were first enacted by the colonial government, there are a lot of uncertainties over their applicability:

> *"Assuming we have arrested the person* [for causing a fire], *we don't know which law we should use to accuse this person. [...] we are so busy looking at parts and parcels of different laws to accuse this person instead of having one legal framework [...] provided by the government".*

Some community members also suggested that these laws were often intentionally overlooked in rural settings since they do not account for everyday subsistence-oriented fire use. This is particularly true where local chiefs who were *"given* [by colonial and post-independence governments] *a lot of power and control over what happens in the rangelands"* turn a blind eye, acknowledging the importance of fire as a cheap and quick land management tool amongst poorer communities.

Challenges for fire management were attributed to a lack of a national legal framework and integrated regional plans specifically dealing with fire governance, limiting investment in management interventions and stagnating conversations over proactive wildfire mitigation measures:

> *"The biggest challenge in fire governance is that the country lacks a legal framework in terms of fire management. What exists is just sectoral policies within small sectors. For example, fire management plans within institutions. It's based on occupational safety and doesn't extend to fire management within landscapes. [...] If a legal framework were to be established, there would also be that aspect of funding fire management action and exploring how other fire challenges can be addressed. Currently, they lack support. This is the challenge".*

Representatives from international wildlife conservation organisations raised concerns over an increase in *"gentlemen's agreements"* or *"ad hoc arrangements"* permitting select individuals entry into protected areas to access resources. One participant reflected on their experience in the Maasai Mara compared to in Tsavo:

> *"Existing laws are somehow conflicting with regards to the community. The Wildlife Act prohibits entry into the park, whether you are grazing livestock, collecting firewood... but the people here in Tsavo tell you that the Maasai are allowed to graze their cows inside the Maasai Mara National Reserve, but we don't allow the same rights for people here to graze inside Tsavo parks [...] There is a need to align these policies so that they are not selective, assisting in avoiding confrontation that arises when people feel that the government favours some people over others".*

Community members noted incidences of bribery and elitism as a means of gaining entry which have exacerbated local grievances and, in turn, intergroup conflict and competition. They considered such acts as examples of political and economic corruption which serve as national and regional barriers to fire management. They also shared their own experiences of entering casual agreements with park managers to assist in firefighting. They expressed their frustration over the lack of a legal framework safeguarding their lives and assets, as well as the expectation for them to assist without any formal incentives. In many ways, the communities have often felt exploited as a cheap labour force [14].

> *"We need a national strategy for managing fires across all the landscapes because I am finding a situation whereby if you allow access into some of the protected areas without*

*control—just allowing people to enter because they wish to utilise or get something out of it, it will degrade that landscape and it might not ever recover. [. . .] There is also a lot of activism activity directed towards us. So, [. . .] I think we have got to be very careful in defining dos and don'ts. If it's a national forest or a national park, they must be kept like that. We cannot let people get in there and start manipulating the resource. [. . .] We should focus on increasing these natural resources on community land, rather than relying on the protected areas".*

When asked what needs to change in fire management, participants agreed that the development of an integrated fire management plan that encompasses different sectors, land tenure arrangements, and ecosystems is of highest priority. This plan can outline resource needs and areas requiring financial investment, including raising greater awareness, the maintenance of management interventions, and the establishment of early warning indicators. Participants agreed that efforts to increase fire management capabilities would be in vain if they did not address the main social and economic challenges that have prevented the implementation of an integrated fire management plan to date [26]:

- *"Lack of awareness over the positive role of fires [. . .] or the benefits of using fire. This is a barrier".* Historically, negative attitudes were reinforced by state and non-state agents [106]. However, a *"loss of traditional ways of life"* and *"the erosion of traditional knowledge"* have created a *"general sentiment among communities that fire is negative and destructive"*, especially when they *"do not see the benefits of the protected areas"*, *"lack power or incentive to maintain the land when it is lost to the government for development projects"*, and *"do not receive compensation when things burn"* [4].

- Intergroup conflict where *"access to resources is denied"*, *"local people and their practices are suppressed"*, and *"the community is not involved in the protected area"*. *"Tensions are increasing between the KWS, KFS, and ranches"* where the production, provisioning, and benefits of natural resources are redirected to the state rather than local people. As a result, *"fire is used as a social act of resistance by local communities within protected area landscapes"*. For example, *"when you have cows, but you don't have access to the protected area, you don't feel like you are part of that protected area. Even when there is fire, you don't care. [. . .] This aspect of denying access to natural resources is changing local attitudes"*.

- Limited livelihood opportunities among subsistence-oriented communities under climate change and population growth. *"You find that people are looking to fulfil their livelihoods"* and *"burn if pasture, fodder, grass, is overgrown in the protected area. [. . .] When grass is overgrown, it's not palatable"*. Local communities are burning *"for economic reasons"*, such *"burning charcoal to get an income, [. . .] honey harvesting, [. . .] and collecting wood"*. *"In Kenya, there is a poverty situation. A lot of people don't have work [..] and the means to put food on the table, so they end up engaging in activities that result in fires—mostly by accident, but sometimes intentional"*. Since they pay taxes, many communities *"feel that the government should provide them with food and some access to protected areas"*.

Acknowledging these existing barriers, participants agreed that an integrated fire management approach would only prove effective if it provided specific guidelines on community involvement and the distribution of resource benefits:

*"Research shows that if you increase the resource benefits to local communities, they are more likely to appreciate the park. [. . .] This can be a way of managing fire".*

One participant shared his personal experience of conflict on a privately owned plantation bordering a national park and community lands:

*"You get local communities around protected areas that are aggrieved because they are not benefiting from that area, just like in our place. There was a time we had multiple issues because the surrounding community were not getting employed [. . .], so they just started burning the farm. [. . .] They have animals to look after, and because the area is fenced, they could not access pasture to graze their animals. This started to create problems. We had to start getting them involved and employed, and when we started including them*

*and educating them, they started to act as our neighbours because they were benefiting from the farm. We still get accidental fires in this area, but this has reduced".*

The role, agency, and bargaining power of Indigenous and local people in a new integrated fire management plan sparked some conflictual debate between participants of different generations. Some younger participants argued that traditional fire practices are degrading and could compromise biodiversity conservation objectives, therefore *"government institutions are required to control community practices and reduce wildfires".* In response, senior participants argued that such wisdom highlighted a decline in intergenerational knowledge transmission over how to effectively use fire in the landscape, largely due to Kenya's long history of fire suppression and increase in conflict-based burning:

*"This demonstrates the erasure of traditional cultures and diversity". "If the government were to exert authority over fire use and ignore local forms of institutional legitimacy and empowerment, local grievances would only increase"* [39].

Younger participants listened carefully to these arguments and were willing to learn from the knowledge of their elders to build a common path for action.

*"The best way to deal with conflict is to bring the community along, laying the foundation. [. . .] If you negotiate a management plan or policy and it is agreed on, this can serve as the basis for equal distribution of resource benefits".*

Participants concluded that responsibility for developing an integrated fire management plan lies ultimately with the government:

*"The government need to come in at a higher level to implement policy that can cascade to all institutions and departments, giving a desk to someone internal in each institution and implementing policy uniformly across them".* This would help address sectoral and institutional differences, such that *"KWS' main objective will still be wildlife, and the KFS' will be forestry"*, *"but there is a common approach to address wildfires across different landscapes".*

Forestry representatives suggested that formal community-based conservation and natural resource management structures could be established to devolve decision-making rights over fire use to Indigenous people and local communities. They highlighted how the current CFA model could be adapted for the fire context, explaining how CFAs were formalised under the Forest Conservation and Management Act [109] (Article 48–52) to enable community members who pay a prescribed membership fee to participate in forest management and benefit from sustainable resource utilisation [110,111]. However, they expressed concern over the existing shortfalls in this model which would need to be addressed in a new fire management plan. This included the KFS's retainment of forest resource ownership rights and the right to withdraw agreements at any time, preventing the decentralisation of management to local communities and their ability to lobby, initiate rural development, and participate in conflict management. Additionally, they noted that CFAs are inadequately funded and over-rely on external support (e.g., NGOs and local governments), often resulting in the introduction of green conditionalities on their communities which limit their role in decision making and impose strict rules over local management practices, including fire use [14,112–114].

*"The best way to manage fires and understand their impact on species is to incorporate local and traditional knowledge. But since communities are losing their traditional way of life, we need to find a certain meeting point. For instance, we can look at traditional institutions and their ways of predicting change, and the transmission of knowledge between groups should be respected. Fire centres should not be isolated between protected areas and communities, they should be central to both. [. . .] We need legal change where communities benefit from the policy. In terms of benefits, now we only look at economic resources and monetary values like water, but we also need that element of social and cultural benefits. Maybe that way, attitudes will change. Maybe we need more projects like REDD+ in Kenya".*

Participants agreed that progressive laws that are developed pluralistically and formally include local communities can result in the co-management of resources and *"incentivise local people to help manage fires in protected areas"*. But such laws *"need to make sure all sectors in the landscape are included, because wildfires affect forests, national parks, and the communities and their farms. A national wildfire policy could bring and invite us all together—across sectors and communities"* [38].

### 3.3.3. Participatory Art

Three artworks were produced during the workshop by an Indigenous artist from the Kamba community, Shadrack Musyoki, including a live painting, summary piece, and participatory mural (Figure 2). These act as an original and powerful way to further understand, reflect on, and communicate issues surrounding wildfire management in Kenya. The live piece, "Mamboleo" (Current Affairs), captures different dimensions of fire, from domestic use to fire in the landscape, to situating Kenyan wildfires as part of a global phenomenon, demonstrating its intrinsic link with other parts of the world. The summary piece, "Chaguo Ni Letu" (The Choice is Ours), represents the artist's final reflections following what he learnt from the workshop. It juxtaposes a healthy environment involving traditional fire practices with a polluted world ravaged by extreme wildfires and drought due to climate change and poor environmental management. The artist's depiction of the role of traditional fire use as part of a healthy future, along with his title for the piece, is a reflection on the significance of culture, voice, and the empowerment of Kenyan people in tackling environmental challenges. The design of the mural, titled "Mwaki" (Fire), was based upon the workshop participants' rich pictures created on the first day of the workshop, ensuring that the participatory art process was not simply an act of painting, but instead nurtured a sense of collective ownership over the artwork. Participation in artistic processes within a workshop setting was a novel experience for participants and was fully embraced; all participants enthusiastically contributed to painting the mural, guided by the artist. This process formed a sense of togetherness and enjoyment, creating a shared memorable point within the workshop that is captured permanently and boldly through the final painting. The artwork is therefore more than a representation of the different dimensions of *"what does fire mean to you?"*, but also a representation of collaboration, togetherness, and shared goals.

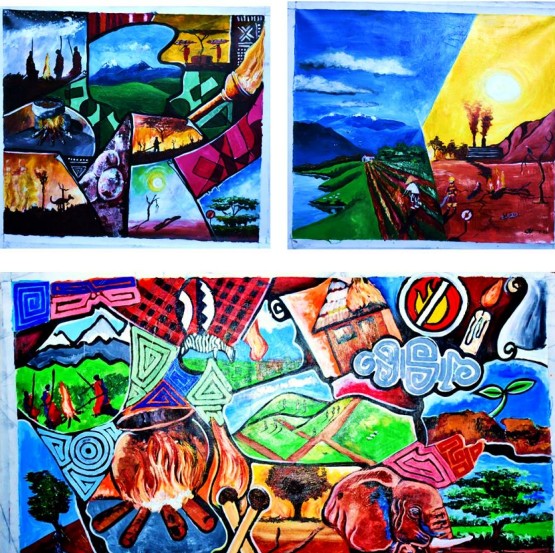

**Figure 2.** Paintings produced in the workshop (acrylic on canvas). Top left: live painting "Mamboleo" (Current Affairs); Top Right: live piece, "Chaguo Ni Letu" (The Choice is Ours)—both by Shadrack Musyoki; Bottom: participatory mural "Mwaki" (Fire)—by Shadrack Musyoki and workshop participants. Original photo: George Munene (Strathmore University).

While participants engaged in participatory art, they were invited to contribute towards a collaborative stakeholder and rightsholder map, mapping their roles, affiliations, the projects they are involved in, other groups they work with, and anyone excluded from their networks (e.g., projects and decision making in their respective areas) (Figure 3). This exercise was to set the scene for the next set of activities focusing on building a road map towards an equitable fire future. Participants gained a better understanding of who was in the room and how, going forward, meaningful collaborations could be developed. They were encouraged to identify existing convergences between people and projects that could be built upon, as well as areas of exclusion, thinking about how and why certain groups are excluded, and how this can be addressed moving forward.

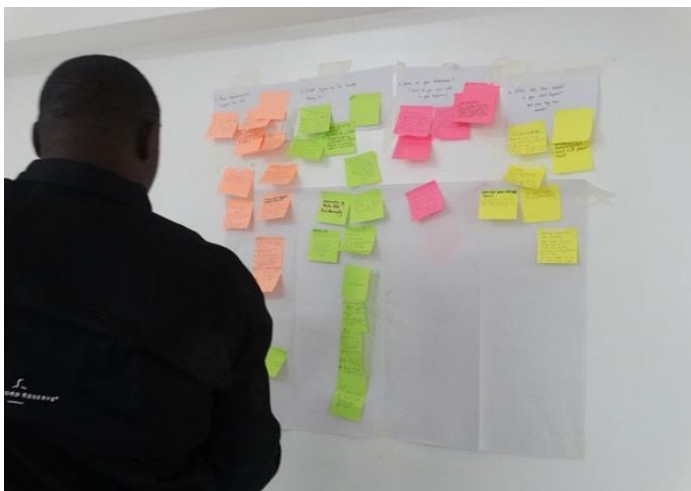

**Figure 3.** Stakeholder mapping exercise. Original photo: Abigail R. Croker.

## 4. London Workshop Activities

Concurrently with the workshop in Nairobi, natural and social science PhD students and postdoctoral and senior researchers from Leverhulme Wildfires met at Royal Holloway University of London to discuss participatory research methodologies and apply them as they heard about the Kenyan context.

Throughout the workshop, London participants connected online with the workshop in Nairobi, watching the same presentations, listening to feedback sessions in Nairobi, and sharing feedback from their own activities. They learned about rich pictures and participatory video as research methodologies and worked in groups to make their own rich pictures (Figure 4) and videos [41]. Their rich pictures demonstrated how perspectives on fire are influenced by researchers' positionalities, knowledge, and experiences. Participants reflected on the extent to which perspectives being presented in Nairobi could be understood by Leverhulme Wildfires researchers.

Participatory video is where a group of people make a video on a topic important to them to drive participatory action and learning through social and political dialogue [60,115,116]. The participatory videos created by the London participants built on their rich pictures. They reflected on the biases and representations in the kinds of data used in research and systems modelling in Leverhulme Wildfires, the agency individual researchers have, the structural limitations placed on researchers (and at different stages of a research career), and on the questions they would like to ask the participants in Nairobi. The workshop concluded by developing a joint declaration to guide future research in Leverhulme Wildfires, including pathways towards (1) recognising a diversity of fire knowledges; (2) making research processes and outputs more accessible and useful; and (3) challenging coloniality and colonialism (Box A1).

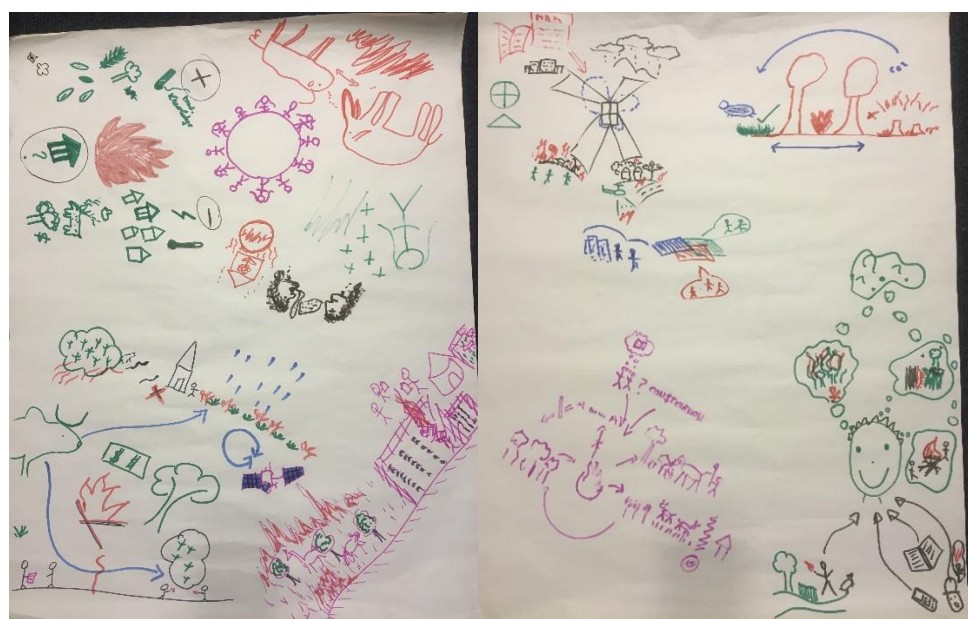

**Figure 4.** Rich pictures created by London Leverhulme Wildfires participants. Original Photo: Cathy Smith.

## 5. Reflections and Next Steps

The experiences and knowledge shared during this workshop by diverse stakeholders, rightsholders, and academics from across Kenya, and researchers and fire practitioners working internationally, were extremely rich and offered novel insights into the significance of intercultural exchange in scientific research and management contexts. The discussions in Kenya revealed the complex challenges and opportunities for more equitable approaches to fire management across contested conservation landscapes. Participants identified the broader Mount Kenya National Park ecosystem, Tsavo and Meru Conservation Areas, and Kirisia Forest in Samburu as fire hotspots and challenging landscapes in terms of park–community relations. The participatory collaborative art was a particularly important aspect of the workshop for participants since it gave them time and space to engage in Indigenous traditions of communication and interpretation. They reflected upon the in-depth discussions held over the two days while revisiting their initial response to "what does fire mean to you?". Participants re-dissected the complex wildfire challenges revealed throughout the workshop by painting the most prominent ideas and relationships. In this process, they discussed their initial responses and how their understanding of fire challenges had evolved over the two days. While the mural displays variation in participants' experiences and attitudes towards fire, the discussion held during this exercise revealed processes of knowledge co-development and desire for collective action, shaped by a confluence of Indigenous and Western understandings of fire [116]. One participant commented that *"the use of arts is an exceptional way of communication that can influence a global audience and is currently underexploited. It facilitated the blending of science with local and Indigenous knowledge to identify the problem and develop measures together, reducing parachute science"*.

Members of the BID-C also expressed the importance of this exercise for providing a *"more contemporary and experiential approach"* which can increase the *"participation of youth [. . .] alongside policymakers [. . .] to raise more awareness on the topics of wildfire governance, wildfire justice, and wildfire management"*. The artworks have also facilitated wider dissemination of the workshop through physical exhibitions and through conference presentations to global audiences. This contributes to broadening awareness and discussions over challenges related to wildfire management in Kenya, as well as in other parts of the world where the issues captured in these artworks, such as evolving Indigenous fire practices and the coloniality–wildfire nexus, are experienced. Regarding the aims of the workshop

to establish research partnerships and explore opportunities for decolonising fire management [117], these artworks have catalysed further art–science collaborations between Leverhulme Wildfires and BID-C relating to climate justice [118]. They have also led to additional projects with Shadrack Musyoki centered around science communications for diverse audiences.

Online engagement with London participants allowed both groups to exchange knowledge and learn more about these challenges in the Kenyan and global context where colonial legacies have shaped contemporary conservation arrangements and fire management. London participants drew similarities between the Kenyan context and Australia, India, the USA, and countries across Central and South America and Sub-Saharan Africa [14,15,20,32,119–123], where they have been researching fire governance challenges [41]. They found the hybrid format to work well, both in terms of saving air-miles, with only one workshop facilitator required to fly to the Nairobi, and in providing an opportunity to listen to the Nairobi participants' perspectives without dominating the conversation, all the while hoping that their contributions offered a different perspective. In London, training activities enabled participants to reflect upon the benefits and limitations of organising intercultural workshops in their own research contexts, as well as assess in real time how far the activities in Nairobi created an equitable environment for participation and assisted in achieving the workshop's core aims:

> *"The shared elements and opportunities for exchange reinforced how we are all implicated in decolonising research, fire management, and international development. At the same time, having separate spaces to consider how we specifically are implicated reinforced that our responsibilities are different, and allowed us to think about concrete steps to take forwards in our contexts".*

Both social and physical fire scientists attended the workshop in London, including those working with big data and using quantitative methods. To date, research incorporating decolonial methodologies has mostly been situated within the social sciences [124,125]. Due to uncertainty regarding the relevance of decolonisation in their research projects, physical scientists proved initially more challenging to engage with. However, given the increasing power of the data economy and technological algorithms over management decisions and policy, fire scientists working with remote sensing data and Earth system models were willing to reflect on how they work with and represent human–fire interactions and their role in promoting an equitable fire future [13,126]. While participation was skewed towards the social scientists, it was promising to see several data scientists engage in this workshop series.

*5.1. Limitations*

> *"There is a danger that readers from an alternative epistemological position will judge the paper in terms of knowledge claims relevant only to their own epistemological position"* [127].

Writing this report allowed us time to reflexively examine and reflect on our individual and group positionalities as researchers, workshop coordinators, and communicators of local perspectives. We encourage readers to also reflect on their own positionality and epistemological dispositions when interpreting the findings presented here, returning to questions of decoloniality in the context of their own research and management practices [49]. We asked ourselves *"to what extent were local fire users and Indigenous people involved in the conception and design of the workshop and in the development of the analysis presented in this paper, and how, if at all, might they have had a more active role in this process?"* and *"how successful was the workshop in sustaining intercultural discussion between Nairobi participants and with London participants, and for building long-term relationships?"*

While the workshop responded to local questions identified through research being carried out across the Tsavo Conservation Area, the design of the workshop and planned activities, online presentations, and the analysis presented in this paper were led by European

and Kenyan researchers educated in Western institutions or affiliated with international research centres and organisations. Despite offering Kenyan participants the option to read and comment on this paper to ensure that their perspectives were accurately presented in this report, we must question the degree to which these findings have been constructed through a Western lens, particularly with regard to our subjective interpretations of individual and collective beliefs and value judgements based on a priori knowledge and assumptions [13]. For instance, gaining feedback from Kenyan participants relied on them having timely access to communication technologies, possibly omitting some of the most marginalised voices in the co-development of this narrative. Feedback was mostly confirmative rather than interactive, perhaps due to the academic format of this paper which might have made it difficult for participants to legitimately engage in this process. More active forms of collaboration in this analysis might have facilitated greater representation in the production of knowledge on fire governance challenges in Kenya [13]. However, it would likely need to be presented in a format that is more accessible and of greater sociocultural relevance to diverse Indigenous and local fire-user communities, rather than the global scientific community. This presents a challenging interface for scientific researchers who are trying to embed principles of decolonisation in their research to promote equitable management and policy outcomes, while also wishing, and sometimes required, to share their findings with the international scientific community in adherence with academic standards in peer review and publication processes. To address some of these challenges, academic outputs can be translated into more accessible formats and translated into local languages. We have published an article in both English and Swahili summarising the workshop activities [41], and we are planning a summary report which will be co-developed with workshop participants, translated, and shared with relevant stakeholders.

The extent to which the workshop fostered meaningful dialogue between London and Nairobi participants was difficult to measure. While online engagement allowed both groups to exchange knowledge, technological limitations and language barriers often made it difficult for participants to hear and understand the discussions. Online engagement can limit opportunities for individuals to explore alternative means of communication, resulting in their points being lost in translation or misinterpreted. Additionally, decolonisation was perceived differently by both groups. Nairobi participants were interested in how decolonial approaches can be practically incorporated into policy action, such as through the creation of community-based associations and the development of a nationwide integrated wildlife management policy. Meanwhile, London participants were focusing on how their research can embed principles of decolonisation to support such action in the contexts they are working. To some extent, we question whether the conversations held in London were relevant to local stakeholders and rightsholders in Kenya, as equitable research is the responsibility of the scientific community, not the researched communities [39,128].

While we acknowledge the limitations on the participatory nature of this workshop, it is also important to acknowledge the challenges embedded within collaborative research paradigms that reinforce hierarchical researcher–participant relations and create biases in participant representation. Greater collaboration with local communities does not automatically mean that principles of decolonisation have been considered in the research process, nor that research outcomes are equitable. For instance, there are often invisible costs imposed on local communities in participatory research activities, such that participation often demands free labour and can be resource and time intensive [50]. In providing transport, accommodation, and meals to all participants, we were generally successful in mitigating some of these costs, and diverse stakeholders and rightsholders attended from across Kenya. However, we realised that time was one of the largest costs to participants in engaging in this workshop, with several invitees, mostly local chiefs and elders, unable to attend due to time constraints [129]. Their responses were positive in that they recognised the importance of this conversation, but they were unable to divert their attention away from dealing with the hardships resulting from a multi-season drought and the imminent threats this posed to local livelihoods and survival. Therefore, participation was skewed

towards representatives from government, non-governmental, and private organisations who were slightly further removed from dealing with daily subsistence-based challenges. Considering this, the perspectives represented in this report are likely biased towards those who hold greater bargaining power in rural affairs, with constructions of community-based approaches reflecting more top-down objectives [113].

These limitations require careful consideration in the future co-creation and co-development of intercultural workshops to foster more equitable participation. Importantly, we need to carefully assess the willingness of local people, and the capacities of both the institution and local people, to work together over long time periods, identifying the resources required to facilitate this.

*5.2. A Road Map to an Equitable Fire Future*

Participants agreed that conservation landscapes need to prioritise the co-development of an integrated fire management plan which encompasses complex, overlapping social–ecological systems (e.g., ecosystems, jurisdictions and governance systems, economic activities) and sectors, while being adaptable to the local context and flexible under future climate change [26]. Representatives from a range of government agencies, organisations, and institutions strongly supported Indigenous and local community discourses over their rights to natural resources, stating that for the plan to be effective and sustainable in the long-term, it must provide clear guidelines for the involvement of Indigenous and local people in decision making and distribution of benefits [102].

Prior to the workshop, Abigail Croker interviewed lawyer and CEO and founder of Bright Tide, Harry Wright, on how to influence policy from the bottom up, *"where every stakeholder and rightsholder 'has a seat at the table' and Indigenous and local people have significant oversight"*. The interview was prerecorded and played to Nairobi participants before they agreed on a common course of action to work towards an integrated fire management policy. The interview revealed two main pathways for bottom-up action, summarised in a leaflet distributed amongst participants for future reference [41]. The first pathway focused on communication flows through multi-level roadmaps and workshops, and the second through legislative inquiries with established roundtables and party groups. Both pathways share a set of key requirements. This includes understanding local, national, and international legal frameworks and multi-lateral directives, prioritising multi-level stakeholder engagement and legitimate local involvement throughout the entire policy process, and building networks of local lawyers operating across different jurisdictions and fields of law [37,38,130].

Participants reflected on this process of policy development, expressing their appreciation to learn about how they can work together to lobby on behalf of their communities, organisations, and institutions. One participant stated that *"the diverse policies governing conservation have failed to address the needs of local people and their rights to access resources in protected areas. [...] Joint with strict fire suppression policies [...], this has led to increasing wildfires in heavily contested protected area landscapes. Policymakers rarely organise dialogues and discussions, [...] exacerbating the problem"*. Participants acknowledged that traditional fire knowledge and practices have been overlooked since the colonial era, stating it is clear that *"land, conservation, and forest management policies were derived from colonial land policies in Kenya, determining how land was and is being allocated for different purposes, such as conservation and development projects. Recent amendments to existing acts and the promulgation of Kenya's constitution in 2010 failed to solve the problem. The discriminatory aspects of colonial policies inherited from the colonial ordinances continue to advance non-consultatory approaches to resource management"*. In light of these injustices embedded in policymaking, participants showed their eagerness to progress with multi-group dialogues *"to enhance the revision of existing policies and formulation of people-centered policies that reflect the current needs of local governments and community and rightsholder groups to restore their confidence in the management of natural resources and emerging issues across conservation landscapes"* [131].

This led to the formation of a Joint Declaration on Wildfire Management in Kenya (Box A2), created and signed by all workshop participants, showing their commitment towards supporting the development of an equitable policy.

*"The joint declaration on wildfire management in Kenya shows the commitments of the stakeholders and rightsholders in actively participating in fire management policy development, managing fires willingly as well as through collaborative research to help build policies that respect indigenous peoples' ideologies, enhance equal rights, and most of all adopt scientifically informed decisions in their respective landscapes".*

To address concerns over the equal participation of Indigenous people and local communities in all stages of the policymaking process and decision making over fire use [37], participants agreed to continue exchanging ideas and engaging in dialogue, pushing for a new integrated and collaborative paradigm in fire management. With this in mind, participants established a fire working group communications channel via WhatsApp to share ideas and plan future action. They identified the need to juggle international, national, and local resource management and conservation objectives, acknowledging the scalar disparities between local socioeconomic systems and policy-relevant institutions, and the evolution of attitudes and practice in response to macro-environmental changes. Ideas on how to address these challenges are ongoing, and this WhatsApp channel provides space for all participants to contribute to the conversation.

The attendance and participation of Indigenous and local people, institutional and organisational representatives, and government agencies were generally successful; participants expressed how useful they found the workshop in bringing groups together who do not normally interact and the importance of intercultural exchange in fire management:

*"The methodologies applied during this workshop were effective in bringing to light the nuances in landscape-related challenges. [. . .] The discussions remained constructive and organised, with no disagreements or heated conversations disrupting the flow of information. [. . .] This workshop provided me with a new perspective on environmental sustainability. Being exposed to the essence of community engagement in wildfire management across contested areas [. . .] is an asset, especially in realising the importance of collaboration between communities and government sectors for the betterment of our ecosystems".*

The success of this workshop was in in part due to the connections the academic organisers had previously created with communities and organisations, largely through being present in the environment, open to learning, and establishing relationships built on trust and reciprocity with local people (Figure 5). These positive relationships between researchers and participants are evidenced on the WhatsApp channel, such that individuals continue to share information to work towards a more equitable fire management approach, as well as engage in more general conversations that are important for building, maintaining, and supporting trusting networks [39]. Long-term collaborative relationships have also been established between Leverhulme Wildfires, the BID-C, researchers working in institutions affiliated with both centres, and local management organisations. For example, through the network established at this workshop, a local participant representing a conservation NGO was offered and has started a fully funded PhD at Leverhulme Wildfires (through King's College London), continuing research on the interactions between wildfire patterns and large mammals across Kenya's protected areas. The development of a partnership with Strathmore University has also led to a recent successful grant application on fire and cultural heritage, involving Leverhulme Wildfires (through Imperial College London), Strathmore University, and other partners, with Kenya as one of the case study countries.

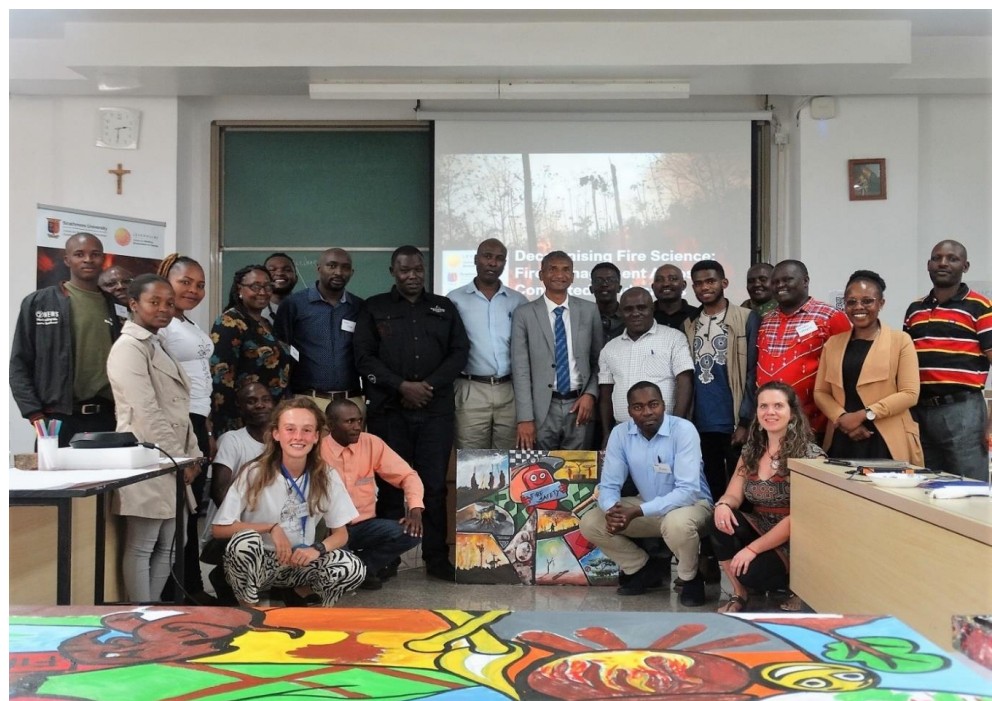

**Figure 5.** Workshop conclusion with participants in Nairobi, Kenya. Original Photo: Abigail R. Croker.

Creating such networks is a critical step towards decolonising our fire science and management. We can begin to address the limitations of participatory research and take seriously the co-creation and co-development of research activities, from project inception to communication. All the while, researchers working in intercultural environments need to reflexively reflect on their positionalities and subjectivities that might impact how, why, and for whom these networks are created [132–134]. This workshop represented diverse social–ecological systems and cultures, incorporating decolonising methodologies that have been practiced in intercultural settings globally. Therefore, the approach and findings presented here can be adapted to other contexts, particularly where fire governance challenges are embedded in a colonial–wildfire nexus.

> *"This event serves as a testament to the importance of collaborative efforts in fostering sustainable solutions for our landscapes"*.

**Author Contributions:** Conceptualisation, A.R.C., A.E.S.F., J.M. and C.S.; Methodology, A.R.C., A.E.S.F., J.M. and C.S.; Formal Analysis, A.R.C., A.E.S.F., J.M. and C.S.; Investigation, A.R.C., A.E.S.F., J.M. and C.S.; Resources, A.R.C., A.E.S.F., A.C.M., J.M. and C.S.; Data Curation, A.R.C., A.E.S.F., J.M. and C.S.; Writing—Original Draft Preparation, A.R.C., A.E.S.F., Y.K., A.C.M., J.M. and C.S.; Writing—Review and Editing, A.R.C., D.C., A.E.S.F., Y.K., A.C.M., J.M., V.M., E.P. and C.S.; Visualisation, A.R.C., A.E.S.F., J.M. and C.S.; Supervision, A.R.C.; Project Administration, A.R.C., A.E.S.F., J.M. and C.S.; Funding Acquisition, A.R.C. and A.E.S.F. All authors have read and agreed to the published version of the manuscript.

**Funding:** This workshop was primarily funded by the Leverhulme Centre for Wildfires, Environment and Society through the Leverhulme Trust, grant number RC-2018-023. The Grantham Institute, Imperial College London also contributed to funding this workshop.

**Institutional Review Board Statement:** This research was approved by the Science Engineering Technology Research Ethics Committee of Imperial College London, UK (22IC7618, 26 April 2022).

**Informed Consent Statement:** Informed consent was obtained from all subjects involved in the study.

**Data Availability Statement:** The original contributions presented in the study are included in the article, further inquiries can be directed to the corresponding author.

**Acknowledgments:** We thank all participants from Kenya that attended the workshop at Strathmore University for sharing their experiences and understandings and their continuing dedication to foster an equitable fire future in Kenya: Naftal Kariuki (Garissa University), Peter Kamau (Karatina University), Charles Kuria (KFS), Kambaki Lalaikipiani (UN FAO), Douglas leboiyare (Kirisia CFA), Alice Lenjo (TTWCA), Ntarisan Lepushirit (Kirisia CFA), Patrick Mbega (Teita Sisal Estate), James Mbuthia (Sheldrick Trust, Kibwezi), Michael Mkala (Taita Wildlife Conservancy), Festus Mburu (Strathmore University), Valentine Mkanyika (Wushumbu Conservancy), Herbert Mwaghesha (Taita Wildlife Conservancy), Zakayo Maina Ngatia (Assistant Chief in Mathira East Subcounty), Amos Chege Muthiuru (African Wildlife Foundation), Koikai Oloitiptip (Amboseli Ecosystem Trust), Kisilu Wambua (Penda Kujua), and Loise (Mt Kenya CFA). We would like to express our gratitude to Vincent Ogutu for warmly welcoming Leverhulme Wildfires and workshop participants to Strathmore University, Sally Archibald, Tercia Strydom, and Leverhulme Wildfires members for sharing their research and practices with us, Harry Wright for his advice on how to influence policy from the bottom up, and George Munene for photographing the artworks.

**Conflicts of Interest:** The authors declare no conflicts of interest.

## Appendix A

**Box A1.** Joint Declaration on Pioneering More Equitable Ways of Making Fire Knowledge.

The following declaration was co-developed by Leverhulme Wildfires researchers at the *"Decolonising Fire Science: Fire Management Across Contested Landscapes"* workshop held between the 1–2 December 2022 in London, running parallel with the workshop in Nairobi:

*We recognise a diversity of fire knowledges that are all legitimate and can inform each other while existing independently, by:*

*1. creating inclusive spaces for knowledge exchange and co-creation*

*2. experimenting with new research processes*

*We want our research processes and outputs to be accessible and useful, by:*

*1. being mindful of why we are doing our research, for whom, and how those people are involved*

*2. using accessible language, including non-English language*

*We have responsibility as individuals to challenge coloniality and colonialism, by:*

*1. recognising the power we hold as researchers and the power embedded in our research tools, methods and outputs*

*2. respecting the people we work with, including research participants and colleagues*

*3. supporting our colleagues so we can grow as a community of researchers*

**Box A2.** Joint Declaration on Wildfire Management in Kenya.

The following declaration was co-developed by 25 stakeholders, rightsholders, and academics from across Kenya at the *"Decolonising Fire Science: Fire Management Across Contested Landscapes"* workshop held between the 1–2 December 2022 in Nairobi:

*Whilst there are existing practices, structures, and legislations for fire management and governance embedded in the current natural resource practices and policies in Kenya;*

*There is need for: greater community involvement, engagement, and empowerment in fire management, and more broadly in natural resource management; for further access to funding, education, training, and resources to aid effective fire management; for better coordination between parties, and improved communication structures, information, and data management; for further knowledge and expertise including scientific, cultural, and indigenous knowledges; for greater understanding on wildfire risk to aid better preparation; and for a transboundary integrated wildfire management policy, providing guidance across different landscapes, tenures, and ecosystems and that includes community guidelines for fire management in their landscape.*

*Therefore, the undersigned declare:*

● *Their intention to support the development of an integrated wildfire management policy, which would include:*

○ *supporting the establishment of locally based fire management committees;*

○ *accelerating research into traditional institutions and knowledges on fire management;*

○ *carrying out community meetings in specific contested landscapes;*

● *And their desire to create a shared space and to continue to work together towards achieving the above intents.*

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
