# Peer review of "Decolonising Fire Science by Reexamining Fire Management across Contested Landscapes: A Workshop Approach"

_fire, doi:10.3390/fire7030094_

Round 1
Reviewer 1 Report
Comments and Suggestions for Authors
Overall the research, use of workshops (Kenya) and connection to researchers (London) is novel and a good approach to make the journal audience more aware of. In being very Kenya focused, there could be some references to similar works/research approaches globally: Aboriginal Australia comes to mind.

Author Response
Dear Reviewer,
Thank you for your comments. Please see the attachment.
Many thanks,
Abigail Croker, on behalf of the authorship team.

Reviewer 2 Report
Comments and Suggestions for Authors
I suggest to reject this manuscript due to the following reasons:
1. It is just a report, which has no scientificity.
2. Throughout the whole manuscript, I cannot find any contribution for the current fire prevention.
Comments on the Quality of English LanguageIt is readable.
Author Response

(The authors gave the same response as above.)

Reviewer 3 Report
Comments and Suggestions for Authors
I applaud the authors for putting this manuscript together. This is very well-written and described. You have provided much for thought and discussion. Best wishes with your future efforts.
Author Response

(The authors gave the same response as above.)

Reviewer 4 Report
Comments and Suggestions for Authors
The paper is well written. Please create a separate 'limitation' section where you include the limitations of this research in detail and the scope of future work.
1. Please explain more about the boarder aspect of this research in the conclusion. How can this approach be used in other parts of the world?
1. Try to provide the novelty of this research at the end of the introduction and conclusion.
Please include examples and references from different parts of the globe.
Author Response

(The authors gave the same response as above.)

Reviewer 5 Report
Comments and Suggestions for Authors
The case report titled “Decolonising Fire Science: Fire Management Across Contested Landscapes. A Workshop Report Summary” tells the experience of a workshop carried out in Nairobi with Kenyan stakeholders and coordinated by British researchers. At the same time, a workshop was occurring in London with researchers. Both groups connected and virtually interacted. The main objective of the Workshop was to establish an inclusive working fire network. The context and methodology for doing this workshop are well-argued and consistent. The article is clear and complete. It is interesting to read about this kind of workshop. I could find some similarities and also some differences with what is happening with fire where I live and work. So, I consider that showing the results of these experiences through scientific journals is useful for researchers to present and discuss current problems including the perspective of local people, as long as these results are not stopped from also being presented in other media that involve the workshop participants and policy-makers.
Author Response

(The authors gave the same response as above.)
